

# Seminar-case learning model improves clinical teaching: a prospective randomized controlled trial

Peiyuan Li[1], Bin Zeng[1], Xuanmin Chen[1], Zhifeng Liu[2] and Jing Yang[1]

[1] Department of Gastroenterology, The First Affiliated Hospital of University of South China, Hengyang, Hunan, China
[2] Department of Otorhinolaryngology, The First Affiliated Hospital of University of South China, Hengyang, Hunan, China

## ABSTRACT

**Purpose.** The purpose of this research was to assess whether the efficacy of the seminar-case learning model is superior to the traditional lecture-based learning model in the gastroenterology curriculum for first-year graduate students.

**Materials & Methods.** This research was a prospective randomized controlled trial that enrolled 92 first-year postgraduate students with a rotation internship in the gastroenterology department. The students were randomly divided into 2 groups and then subjected to an identical version of the curriculum for 8 weeks. The experimental group ($n = 50$) used the seminar-case learning model, while the control group ($n = 42$) used the traditional lecture-based learning model. Examinations consisted of a theoretical test and a case analysis test, and anonymous questionnaires were used to assess teaching quality.

**Results.** All participants completed the examinations and questionnaires. The average theoretical test score of the experimental group was no statistical significance with that of the control group ($P = 0.17$). The average case analysis test score of the experimental group was significantly higher than that of the control group ($P < 0.05$). The indicators of the experimental group's feedback were better than those of the control group, such that there were significantly higher learning interest and motivation, a better understanding of diseases and knowledge, improvements in clinical thinking and summary ability, and an active classroom atmosphere in the experimental group ($P < 0.05$). However, students in the experimental group felt more burdensome.

**Conclusion.** Compared to the traditional method, the seminar-case learning model showed a higher efficacy. The seminar-case learning model effectively improved students' outcomes and satisfaction, which helped students narrow the gap between theoretical knowledge and clinical practical application.

# INTRODUCTION

With the diversification of education models, advanced medical education with medical students is facing increasing challenges. Traditional lecture-based learning (LBL) is mainly taught by teachers through "lecture teaching". Teachers impart medical knowledge from

Corresponding authors
Zhifeng Liu, liuzf@usc.edu.cn
Jing Yang, yangjing@usc.edu.cn

books to students through a monotonous lecture teaching model, which fails to cultivate independent thinking and practical applications of students (*Estai & Bunt, 2016*; *Zeng et al., 2020*). However, most patients have many symptoms with complex conditions and require doctors with clinical practice abilities (*Karle, 2006*; *Xiao et al., 2007*). Therefore, it is urgent to expand new teaching models and methods to improve the efficacy of clinical teaching.

According to the ICAP framework by *Chi & Wylie (2014)*, learning will increase from passive to active to constructive to interactive. Learning efficacy is enhanced when students interactively engage in discussions among groups (*Chi, Kang & Yaghmourian, 2017*). A seminar, an effective tool to stimulate discussions, is defined as a class or meeting with an intense exchange of ideas on a particular issue (*Skeff et al., 1986*). As the center of the learning environment, students can actively analyze clinical problems under the guidance of teachers. The communication between teachers and students enables multiangle interaction to achieve the harmonious unification of "teaching" and "learning" (*Spruijt et al., 2012*; *Zeng et al., 2020*). Case-based learning (CBL) is based on real case scenarios. Teachers provide real cases to arouse the interest of students in learning and to develop the clinical reasoning of the students (*Ali et al., 2018*; *Liu et al., 2020*). Thus, teachers can help students narrow the gap between theory and practice (*Radomski & Russell, 2010*; *Thistlethwaite et al., 2012*). Case-based learning includes sufficient information and detail to induce active analysis by students, which can improve clinical reasoning skills (*Klein et al., 2019*; *Weidenbusch et al., 2019*).

However, a systematic review of the effectiveness of case-based learning in health education showed there were few large-sample randomized trials with outcomes of empirical data rather than description (*Thistlethwaite et al., 2012*). Seminars were well accepted by medical students in evidence-based medicine learning and might lead to an increase in knowledge, interestingly with a good effect in transferring knowledge into a paper case scenario (*Weberschock et al., 2005*).

The seminar-case learning model innovatively integrates the efficient communications of seminar learning and clinical thinking of case-based learning. We hypothesize the seminar-case learning model can improve efficacy of clinical teaching and satisfy students compared to traditional lecture-based learning. There are few large-sample randomized synthesis trials of the two methods in clinical teaching. We may provide data by prospective evaluation and randomized experimental design.

The purpose of this study was to assess whether the efficacy of the seminar-case learning model was superior to that of the traditional lecture-based learning model among first-year postgraduate students in the gastroenterology curriculum. We performed a randomized study to compare seminar-case learning with traditional lecture-based learning.

## MATERIALS & METHODS

### Participation and groups

Based on the inclusion criteria, the trial enrolled 92 first-year postgraduate students specializing in clinical internal medicine with a rotation internship in the gastroenterology
department of The First Affiliated Hospital of University of South China from January 2019 to December 2019. Oral informed consent was obtained from all students. Decisions about whether to participate did not influence their grades. All participants had already been granted an undergraduate degree from a medical university. All participants were randomly divided into an experimental group (50 students) and a control group (42 students). Our study was approved by The Ethics Committee of the First Affiliated Hospital of University of South China (No. NHFY201973).

## Design

The curriculum contents of the experimental group and the control group were identical and included common diseases in the gastroenterology section of the 8th edition of the Internal Medicine textbook, including gastroesophageal reflux disease, peptic ulcer, intestinal tuberculosis, inflammatory bowel disease, functional gastrointestinal disorder, liver cirrhosis, acute pancreatitis, and gastrointestinal hemorrhage bleeding, which were lectured by two leading teachers separately. The experimental groups used the seminar-case learning model, while the control groups used the traditional lecture model. These two groups also underwent 8 weeks of rotation study in the gastroenterology department simultaneously. We arranged the curriculum of the experimental group and the control group on different days once a week (Monday and Thursday, respectively). The students in the experimental group and the control group were also arranged in different wards ground (two floors of wards) to avoid contamination between groups.

The experimental group used the seminar-case learning model as follows:

### Case selection

A vice chief physician worked as the lead teacher, and a resident physician served as the assistant teacher. According to a specific disease of the internal medicine textbook outline, the leading teacher selected a typical patient hospitalized in the gastroenterology department as the teaching case. The assistant teacher liaised with the patients and obtained their permission 3 days before the class. With the agreement of the selected patients, the assistant teacher sent the patient's anonymized information, including their chief history, daily activities, past history, and results of clinical examinations, to students in a newly established WeChat group. The leading teacher raised some questions about the disease's diagnosis and treatments in advance.

### Preparation work

Students were expected to collect the relevant literature and the latest guidelines based on the clinical data of the selected case. Students were divided into groups of 4-5 people before class. Each group organized materials and prepared answers to questions.

### Seminar-case learning model

The assistant teacher helped the leading teacher provide the learning material and teaching equipment for the experiment group. First, the leading teacher gave a brief lecture (10-15 min) to illustrate the main points of disease, of which the content was a simplified version of traditional teaching. Then, the leading teacher introduced the selected case.
In a seminar, students summarized the disease characteristics and analyzed the results of patients' clinical auxiliary examinations in groups. Students were required to answer preview questions. During this course, the leading teacher discussed with grouped students freely and corrected their answers. Finally, the leading teacher summarized the clinical characteristics of the case and extended the case to the disease, and shared experience with the disease. PowerPoint was used to show the context of the lecture and anonymous patient information as well as the materials of the course. The whole teaching time of each course was 90 min once a week.

The control group used traditional lecture learning as follows:

The learning material and teaching equipment for the control group were also provided by the assistant teacher. The same leading teacher from the experimental group gave one lecture by PowerPoint based on contents of the gastroenterology section of the 8th edition Internal Medicine Textbook and shared experiences with the disease, but with no discussion or case. The leading teacher proposed the same questions and analyzed and answered these questions in the class. The teacher also answered the students' questions after class. The whole teaching time of each course was 90 min once a week, the same as the experiment group. We controlled for the potential variable factors in both the experimental and control groups by the assistant teacher, including the same setting, video assistance, and a corresponding simplified PowerPoint of the teaching context for the experimental group.

## Assessment of teaching quality

After 8 weeks of rotation practice, the experimental group and the control group underwent the same examinations and responded to an anonymous questionnaire at the same time. The examinations included a theoretical examination and a case analysis examination with a total score of 100 points. All test papers and questionnaires were prepared, graded, and recorded by the teaching supervisor of Gastroenterology.

### Theoretical examination

The regular theoretical examination includes 5 questions: 1. What are the common causes of gastrointestinal bleeding? 2. What are the clinical manifestations of decompensated liver cirrhosis? 3. What are the treatments for peptic ulcers? 4. What are the diagnostic criteria for ulcerative colitis? 5. What are the diagnostic criteria for acute pancreatitis? The total score was 100 points, with 20 points for each question.

### Case analysis

Two new cases were presented to the students in test papers. Students were required to answer the key points of the diagnosis and treatments of the disease related to the case in a written form. The total score was 100 points, with 50 points for each case.

### Questionnaire

The questionnaire included nine items on students' feelings and perceptions of their classes. Students filled in a table with a "yes" or "no" after each item in the questionnaire, depending on their perceptions of whether the class had strengthened their various abilities and their fondness of the class.
## Statistical analysis

SPSS 24.0 statistical software was used for data input and statistical analysis. Statistical graphics were completed by GraphPad Prism 8.2.0. The measurement data were expressed as the mean ± standard deviation ($\overline{X}\pm S$). The normal distribution of the data was assessed by the Kolmogorov–Smirnov test (K-S test). If the data were normally distributed, the independent samples $t$-test was used to compare the experimental group and the control group; if the data were not normally distributed, the Mann–Whitney rank-sum test was used. The categorical data were analyzed by Pearson's chi-square test to compare the difference in gender and the students' opinions about the teaching methods in two groups. $P < 0.05$ indicated statistical significance.

## RESULTS

All 92 participants underwent examinations after 8 weeks of rotation. A total of 50 students were in the experimental group, including 23 males and 27 females. A total of 42 students were in the control group, including 20 males and 22 females. There was no statistically significant difference in gender, age, or entrance exam score between the two groups (Table 1).

The scores of theoretical test and case analysis of all participants are shown in Fig. 1. The average theoretical test score of the experimental group was no statistical significance with that of the control group. The case analysis score of the experimental group was significantly higher than that of the control group (Table 2). Forty-two people in the control group and 50 people in the experimental group completed the anonymous questionnaire, and a total of 92 questionnaires were received. The indicators of the experimental group's teaching effect were better than those of the control group, such that there were significant increases in learning interest and motivation, a better understanding of diseases and knowledge, an improvement in clinical thinking and summary ability, and an active classroom atmosphere ($P < 0.05$). However, some negative learning experiences were reported. Some students thought a seminar-case learning model class had taken up too much spare time and led to stress, which weighted the gains on balance. The survey showed that the majority of students hoped to adopt a seminar-case learning model (Table 3).

## DISCUSSION

Traditional teaching is teacher-centered lecture-based learning, which emphasizes the delivery of syllabus and concepts (*Barrett, Yates & McColl, 2015*). Clinical teaching is usually a retelling of the theoretical content of medical textbooks and ignores the cultivation of students' clinical thinking as well as the practical application of clinical theoretical knowledge to some extent (*Singh et al., 2017*). Students have poor enthusiasm for dull theoretical knowledge in lecture-based learning (*Mahler, Großschedl & Harms, 2018*). Due to patients' diverse clinical symptoms and complicated conditions in realistic cases, the traditional medical teaching model cannot satisfy the practical training needs of medical students (*Cleland, 2018*; *Formenti et al., 2015*; *Schmidt & Mamede, 2015*).

**Table 1 Comparison of general data between two groups of students.**

| ☐ Group | Sex | | Age/year | Entrance exam score |
|---|---|---|---|---|
| | Male | Female | | |
| Control group | 15 | 27 | 24.48 ± 1.93 | 79.43 |
| Experimental group | 22 | 28 | 24.38 ± 1.91 | 80.6 |
| t value/$\chi^2$ | 0.3527 | | 0.2390 | 0.7800 |
| p value | 0.5526 | ☐ | 0.8116 | 0.4363 |

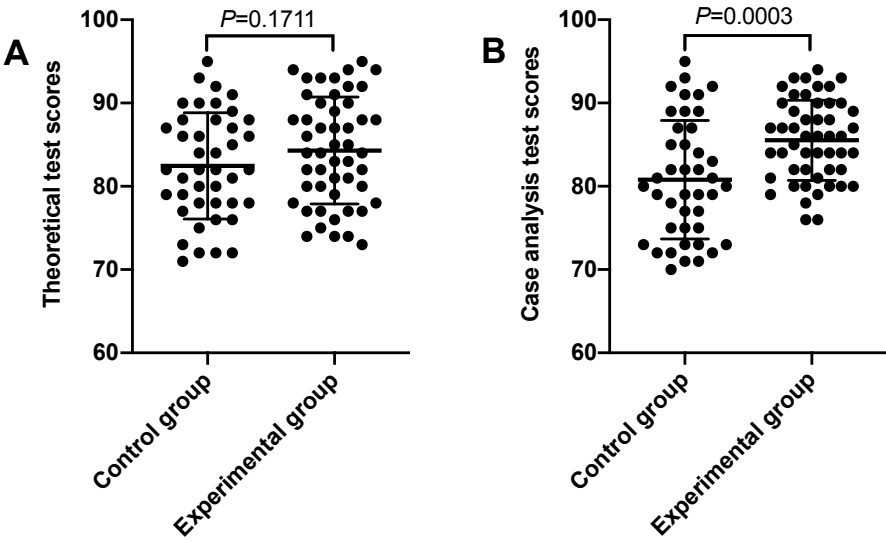

**Figure 1 Theoretical test scores and case analysis test scores of the two groups.** (A) Theoretical test scores of the two groups; (B) case analysis test scores of the two groups (the control group: $n = 42$; the experimental group: $n = 50$).

**Table 2 Comparison of average scores between the two groups ($\bar{x}$ ± s).**

| ☐ | Control group | Experimental group | | ☐ |
|---|---|---|---|---|
| | $n = 42$ | $n = 50$ | t value | p-value |
| Theoretical test scores | 82.45 ± 6.38 | 84.3 ± 6.42 | 1.3800 | 0.1711 |
| Case analysis test scores | 80.79 ± 7.12 | 85.52 ± 4.82 | 3.7,820 | 0.0,003 |

    In our study, the seminar-case learning model that organically combines theoretical knowledge and clinical practice requirements and fully initiates subjective students, are more effective and favored than the traditional teaching model. The students' case analysis performance was improved by the seminar-case learning model, and satisfactory feelings about the class were improved according to the questionnaire. The improved learning effect may have occurred due to the following reasons. First, students are active and self-learning in the preparation work, including browsing the literature online and previewing knowledge. Second, in seminar discussions, the exchange of ideas between the groups deepens the understanding of clinical issues and promotes the full activation of

**Table 3 Comparison of questionnaire results between the two groups.**

| Items surveyed | Experimental group | | Control group | | $\chi^2$ | p value |
|---|---|---|---|---|---|---|
| | n = 50 | | n = 42 | | | |
| | Yes | No | Yes | No | | |
| Increase learning interest and motivation | 37 | 13 | 17 | 25 | 9.2436 | 0.0024 |
| Better understanding of diseases and knowledge | 36 | 14 | 18 | 24 | 6.8395 | 0.0089 |
| Improve communication and expression skills | 28 | 22 | 15 | 27 | 3.0024 | 0.0831 |
| Develop teamwork ability | 26 | 24 | 16 | 26 | 1.2625 | 0.2612 |
| Improve self-learning ability | 30 | 20 | 16 | 26 | 3.5486 | 0.0596 |
| Cultivate clinical thinking and summary ability | 38 | 12 | 10 | 32 | 22.8693 | <0.0001 |
| Activate the classroom atmosphere | 39 | 11 | 11 | 31 | 22.6509 | <0.0001 |
| Occupy time and make a burden | 37 | 13 | 15 | 27 | 12.1016 | 0.0005 |
| Continue to adopt this teaching method | 35 | 15 | 14 | 28 | 10.8989 | 0.0010 |

knowledge structures such as divergent thinking and critical thinking due to the questions between teachers and students. Seminars, as a powerful learning environment, improve students' ability to diagnose and analyze diseases. Third, through the analysis of the selected real cases of patients, students simulated and participated in the entire medical process of patients, taking patient symptoms and signs as a starting point, obtaining specific clinical examination results, and carrying out diagnosis and treatment, thus promoting the practical application of theoretical knowledge. Furthermore, clinical thinking was cultivated in the process of seminar discussion and case learning. Notably, the efficacy of teaching outcomes is higher in the study group, considering an equal amount of hours invested by the teachers, costs, the equivalence of equipment and room space.

Our research found some interesting negative results. Although the average score of the experimental group is higher than that of the control group, there was no statistically significant difference in the theoretical test score between the two groups, which may be explained by self-learning and textbook review after class. Furthermore, some students thought seminar-case learning model occupied more time and was made a burden by the preparation work of relevant literature and the latest guidelines of the relative case. This negative learning experience may be improved by choosing relatively simpler cases and simplifying the preparation work. Comfortingly, 70%(35/50) of the experimental group tended to the seminar-case learning model.

A seminar is a class at a college or university in which the teacher and a small group of students discuss a topic interactively (*Runquist et al., 2006*). In previous research, the seminar method has been shown to an effective and feasible way to improve clinical teaching (*Skeff et al., 1986*; *Takata et al., 2013*). Seminar-based teaching greatly increases students' learning motivation. The mutual communications between teachers and students not only consolidate theoretical knowledge but also expands students' horizons (*Landry et al., 1994*). Moreover, seminars turn the "lecture style" into a "discussion style", making the teaching atmosphere lively and relaxed, democratic, and equal, thus increasing students' enthusiasm and intention to learn (*Spruijt et al., 2012*).

Case-based learning combines clinical theoretical knowledge with real patient cases (*Dickinson et al., 2018*; *Thistlethwaite et al., 2012*). In our study, students needed to find clues from the limited information of the cases and finally make the diagnosis and treatment plan based on the patient's symptoms, signs, and auxiliary examinations in the real case. In the process of simulated diagnosis and treatment, teachers encouraged students to think logically and critically to put clinical theoretical knowledge into practice (*Edelbring et al., 2012a*; *Edelbring et al., 2012b*; *Weidenbusch et al., 2019*). Similarly, students' ability to analyze and solve clinical problems was fully cultivated to better apply their theoretical knowledge to clinical use (*Dickinson et al., 2018*; *Liu et al., 2020*).

In our research, the seminar-case learning model made full use of its advantages in teaching and achieved better effects than the traditional model. However, there were still some limitations in this prospective study. First, although the course content was the same between the two groups, there were minor differences in teaching PowerPoint slides. We were not able to determine whether these minor differences influenced the results. Moreover, courses in both groups were introduced by one lead teacher to eliminate bias due to teaching level. This means that double-blinding was not possible, which may have affected the validity of the findings. Notably, students in the experimental group were likely to spend more time studying after class, which can not be accurately counted in the experimental design and may result in a deviation in efficacy. The superiority of the seminar-case learning model should be supported by more randomized controlled data from diverse departments in multiple teaching hospitals.

## CONCLUSIONS

In general, our study applies the seminar-case learning model to clinical teaching in the gastroenterology department. Compared to the traditional method, the seminar-case learning model showed a higher efficacy, which demonstrated better outcomes and feedback compared to the traditional method. The seminar-case learning model combined lecture teaching and discussion based on real cases, realizing the integration of theoretical knowledge and clinical practical application and exerting a profound impact on medical education. The seminar-case learning model, as an effective method for high-quality education, can be adopted by educators.

## ACKNOWLEDGEMENTS

We would like to thank our colleagues in the graduate administration office and the medical teaching and research department for administrative and academic support. We would also like to thank all postgraduates enrolled in this study.

### Funding

This work was supported by the Education Reform Research Project of University of South China (NO.2017XZG-YY34) and the Health and Family Planning Commission of Hunan

Province (NO.20200532). The funders had no role in study design, data collection and analysis, decision to publish, or preparation of the manuscript.

## Grant Disclosures

The following grant information was disclosed by the authors:
Education Reform Research Project of University of South China: NO.2017XZG-YY34.
Health and Family Planning Commission of Hunan Province: 20200532.

## Competing Interests

The authors declare there are no competing interests.

## Author Contributions

- Peiyuan Li conceived and designed the experiments, performed the experiments, prepared figures and/or tables, and approved the final draft.
- Bin Zeng and Xuanmin Chen performed the experiments, prepared figures and/or tables, and approved the final draft.
- Zhifeng Liu conceived and designed the experiments, analyzed the data, prepared figures and/or tables, authored or reviewed drafts of the paper, and approved the final draft.
- Jing Yang conceived and designed the experiments, performed the experiments, analyzed the data, prepared figures and/or tables, authored or reviewed drafts of the paper, and approved the final draft.

## Human Ethics

The following information was supplied relating to ethical approvals (i.e., approving body and any reference numbers):

The Ethics Committee of the First Affiliated Hospital of University of South China granted Ethical approval to carry out the study within its facilities (Ethical Application Ref: NHFY201973).

## Ethics

The following information was supplied relating to ethical approvals (i.e., approving body and any reference numbers):

This study was approved by The Ethics Committee of the First Affiliated Hospital of University of South China (No. NHFY201973).

## Data Availability

The ages, gender and scores of the students are available in the Supplementary File.

## Supplemental Information

Supplemental information for this article can be found online at http://dx.doi.org/10.7717/peerj.11487#supplemental-information.

![PeerJ]

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
