# Peer review of "Seminar-case learning model improves clinical teaching: a prospective randomized controlled trial"

_PeerJ, doi:10.7717/peerj.11487_

## Round 0.1 · original submission · Major Revisions

Thank you for submitting this manuscript. It has been reviewed by three experienced and well-respected scholars in the field of medical education. Should you choose to revise and resubmit, please pay careful attention to traditional formatting for research papers including the provision of hypothesis, aims, and related research questions.

Also please address all of the reviewers' concerns specific to traditional and accepted review of and provision of ethical research protocols and informed consent.

Reviewer 1 ·

Basic reporting

The intention of the authors is to evaluate the use of seminar-based teaching as compared to the more traditional lecture-based teaching. To no surprise, they find that the seminar-based method provides may advantages. This is a common finding and, sadly enough, this study does not contribute new knowledge on the topic.
Overall, the study lacks in rigour in methods (lack of validated measures) and in the reporting. See comments below.

The introduction is poor and the literature review is shallow.
Statements meant to provide a rationale for the study is poorly described and backed up only with editorials and “perspective” articles (Formenti et al. & Armstrong et al.,)
The first sentence states challenges of “the network information era” and diversification of educational models as a problem with reference to a editorial journal comment. Why are these problems?
“Nowadays, clinical medical students have poor clinical application ability of medical knowledge (Armstrong et al. 2018).”

The author claims that the origin of “seminar” is German and refers to Skeff et al., 1986. I disagree, I would say it orginates from the latin “Seminarium” and I cannot find that the origin is mentioned in the Skeff et al. paper.

The authors mix “seminar” as a method with the method “Case-based learning” and their own "Seminar-case learning model". This makes it confusing and not convincing. Overall, the study lacks anchoring in clearly defined concepts and conceptual framework. Thus, the results of the comparison cannot contribute to new knowledge.


Table 3 has wrong labels on experimental and control groups which is detrimental to the discussion and conclusions. (based on the presented numbers and the text “42 people in the control group and 50 people in the experimental group completed the anonymous questionnaire”)

Experimental design

Methods
The outcome measures (theoretical examination and the case analysis examination) are poorly described and not validated in other research.

Validity of the findings

Results
The authors state that “Both the average scores of the theoretical test and the case analysis test of the experimental group are higher than the average scores of the control group”
This is not true since the p value for the comparison of the theoretical test is not below the selected critical value of 0.05.


Discussion
Also in the discussion the authors refer to improvements in the theoretical test without a significant p-value: “The results showed that not only the students' theoretical performance was improved by the seminar-case learning model,”.

Conclusions
Solid conclusions cannot be drawn because outcome measures are poorly described and not anchored in a thorough conceptual framework.

Reviewer 2 ·

Basic reporting

Basic reporting
• English is clear and easy to understand.
Please give space before the bracket of a reference in the in-text citation.

• In the introduction, the justification to test the seminar-case learning model in an RCT
is not adequately presented. Authors should revisit the article from Thistlethwaite et al. 2012 (The systematic review of the effectiveness of case-based learning in health
professional education. A BEME systematic review: BEME Guide No. 23) and use this article to provide evidence on literature gaps on Case-Based Learning. The authors should be able to show in the introduction how this study differed from others and how the result will add to the body of knowledge.

. Authors should have a clear research question and hypothesis at the end of the introduction section.


• Some sentences were not in the correct sections of the article. For example, in the last paragraph of the introduction, the authors reported methods and findings. In the section of participation and groups, the authors mentioned information about participants and statistical analysis of participants’ characteristics, that should be moved to the result section. Please check throughout the manuscript to ensure that information are written in correct sections of the article.

Experimental design

• Authors requested oral consent from the students. I wonder if this approach has been approved by the ethical committee. It is also not clear if students have obtained information that their decision for participation or nonparticipation will influence their grades. The authors should also explain how they will ensure that students in the control group does not get less benefit in learning compared to the experimental group.
• This experiment was conducted in a few weeks. Students in both groups may interact with each other. The authors should discuss the possibility of contamination between groups.
• Courses in both groups are introduced by one lead author. Blinding is not possible. The authors should discuss this issue and how it can influence the result of the experiment.

Validity of the findings

• Authors should report how the instruments to measure the outcome were developed and validated. I understand that there are two measurements: the theoretical examination and a questionnaire. The validity and reliability of the instruments should be discussed.
• The authors found no statistical difference in the theoretical test between two groups. This is not in-line with the references that authors use as a basis for the choice of seminar-case learning model. Authors should be able to explain this phenomenon. It is not enough to state that as the average score of of the experiment group is higher means that this model can promote understanding of the theoretical knowledge.

Additional comments

• There are a lot of studies on innovation in medical education to nurture clinical reasoning (or authors mention in as clinical thinking). Active learning has also been advocated for years as a preference over lecture. I cannot see why it is important to compare both methods.
• Nevertheless, this is a nice report of innovation in medical education. The authors introduced a case-based learning approach, that they named ‘seminar-case learning model’ for graduate students. Please consider reporting this innovation as a case study in education, rather than as original research. Or, authors can enrich the data using some qualitative research approaches such as interviews or focus group discussion, to explore more understanding about how students learned using this approach and how the practice implication was in the work-based learning

·

Basic reporting

Although the language is mostly appropriate, there are quite a few errors in the English used, both errors in grammar as in style. I would recommend the authors to ask a native(ish) speaker to review the text to improve its style.

The number and depth of the references provided appears to be adequate and sufficient.

The raw data is shared, including relevant figures and tables. The structure of the article is concise and professional.

The article is self contained with a relevant result, but strictly speaking, there was no central hypothesis formulated. Instead, the conclusion is stated in the introduction ("The seminar-case learning model showed significantly superior to the
65 traditional teaching method.") I would recommend leaving this out, and replacing it by a hypothesis. This hypothesis is suggested in both the abstract and the introduction, but never officially stated.

Experimental design

The research is within the Aims and Scope of the journal.

The research question is not very well defined, although it is relevant and meaningful. The purpose is stated, but a clear cut central hypothesis is not formulated. Therefore, the endpoints are not clear and the interpretation a bit questionable. Is higher test scores the proof of superiority, or the students' satisfaction? Moreover, the end results are presented as being (statistically) significant, which is only true for the 'Case analysis test score'. But is a <5 point difference on a 100 point scale indeed 'clinically' or scholarly significant? A more precise formulation beforehand would improve this interpretation. Would this be a clinical trial, this would probably proof a 'non inferiority' at best.

The investigation itself throws up some questions:
1. The ethical approval is in Chinese. I trust the authors to be writing the thruth, but I can't check that.
2. The approval of the test subjects was oral, not written, and thus can't be checked. In this specific context it is perhaps acceptable, but stil, something to frown upon.
3. Patient data is shared through WeChat. As I do not live in China, I can't check this, but in Western Europe, sharing of these data through a non-encrypted AND medical-grade App, is a breach of law. Specific medical Apps should have been used for this in Western Europe (and I suspect, the US). This is not negative to the research itself, but if this is accepted and/or standard practice, I would recommend stating this specifically in the text.
4. The specific test used on the students is unclear. Apparently, both groups received the same test? Is this a standard test used? Was is a test specifically designed and/or made for this research? If so, the risk for bias exists, if the test is more focused on clinical reasoning or details shared during the seminars. Especially if the test is written AFTER the seminars. Elaboration on this point is advised.
5. I cannot infer from the text whether the group size in which the seminars and lectures are given is the same. If not, this could induce bias. Elaboration on this part, or dicussion of this in the Discussion would be helpful.
6. I have the suspicion that the results of the Questionnaire is wrong, as the results showed suggest less 'Better understanding diseases and knowledge'. If it is correct as presented, please discuss this in the article.

Validity of the findings

I have not replicated the statistical analysis, but as such, looking at the data, it looks plausible, so I have no reason to doubt the conclusion.
My interpretation is that the data is robust, and sound.

My main point of criticism is, as stated before, lack of formal cut-off and/or end-points formulated beforehand.

The conclusion, that seminar style teaching is superior is therefore a bit strong, althoug a suggestion of possible superiority is present. The wording used, 'more efficient', can only be concluded if the total amount of hours invested (both from students and faculty) is accounted for. This, however, is unknown, or at least not mentioned in the article.

Additional comments

It is an interesting read, and could be an important read as well. Although there is need for some major revisions (mostly due to the formulation of the Original hypothesis), I think it is worth the effort to make these revisions.
Also, I repeat my recommendation for a thorough English language review. I can understand that it must be (very) hard for a Chinese native speaker to write in English, for which I extend my praise. But some improvement is required for definitive publication.

---

## Round 0.2 · Major Revisions

Thank you for your continued persistence as it relates to the reviewing process. The revision process is iterative and I hope you will take the time to attend to the the reviewers concerns as outlined by the reviewers. I do think it will be important to attend reviewer number 3's concerns. It may be that you would need to acknowledge these concerns in the limitations section of the manuscript.

Reviewer 2 ·

Basic reporting

Authors need to check some references as attached in the comments in the pdf file.

Experimental design

The research question is now also much more clear than the previous one. More supports are needed to show how research fills an identified knowledge gap.
Some issues in the method section need elaboration. Please refer to the comment in the pdf files.

Validity of the findings

This part is very much improved.

Additional comments

Overall, the manuscript is very much improved. Please kindly refer to some comments in the pdf file.

Annotated reviews are not available for download in order to protect the identity of reviewers who chose to remain anonymous.

·

Basic reporting

The English used has markedly improved.
The structure, literature references are up to standard.
In my view, the study still lacks a well-defined hypothesis, and therefore makes the central conclusion a lot less sturdy

Experimental design

The lack of a central hypothesis makes the design of the experiment a bit shoddy.
After all, what is the reason for this experiment? A higher score on the test (the main outcome), a happier student? Something else? Is efficacy something to consider?
I still think that a well defined central hypothesis would greatly increase the study's value.

Validity of the findings

Within the scope of a preliminary study, the validity is good. The statistics seem solid.
But a 'hard' conclusion is lacking. Due to the lack of a central hypothesis, even a non-conclusive can't be stated. That is too bad in my view. I'm convinced that with a different starting point, a better conclusion could be reached.

Additional comments

The reason that I state 'major revisions', is due to the fact that the central hypothesis is a fundamental part of the paper. Therefore, the term 'minor revisions' seems inappropriate.

---

## Round 0.3 · Minor Revisions

Dr. Yang, I want to thank you for your patience working with us through the review process. Your revised manuscript has been reviewed by one of my esteemed colleague's who provided review of your initial submission. Please see their comments and recommendations for this iteration of your manuscript.

·

Basic reporting

No comments. The English is high standard, as was the previous version (as compared to the first).
Good structure and context provided.

In the PDF, the word efficacy is damaged somehow. I'm not sure if that was caused by the PDF generation, as the Word document is OK.

Experimental design

The presence of a clear description of a central hypothesis makes the article a lot more impactful in my view. Especially compared to the previous version.

The only advice in this context is, perhaps, to provide a bit more insight in efficacy:
As the students in the study group state, they experience (a bit) more stress. So, apparently, they put in more hours. This could be seen as less efficacious.
On the other hand, from a university (of even patient's) view, the effect from an equal amount of hours invested by the teachers, seems to be a bit higher in the study group. The costs are (usually) on the side of room-space and teacher's hours, so this is a higher efficacy on that part. Some remarks on this part would elucidate perhaps a bit more.

Perhaps this comment is confusing. I hope you understand my points.

Validity of the findings

The conclusion seems reasonable and correct.

Additional comments

None, besides compliments for the improvements in the article.

---

## Round 0.4 · accepted · Accept

Dr. Yang, I want to personally thank you for your patience with the review process. Writing for academic publication is by its nature iterative and I am delighted to see this manuscript move forward.

·

Basic reporting

Up to standard, same as before

Experimental design

Up to standards, and withing aims and scope of PeerJ.

Validity of the findings

No further comments.

Additional comments

None